# Potential Original Drug for Aspergillosis: In Vitro and In Vivo Effects of 1-N,N-Dimethylamino-5-Isocyanonaphthalene (DIMICAN) on *Aspergillus fumigatus*

**DOI:** 10.3390/jof8100985

**Published:** 2022-09-20

**Authors:** Zsuzsa Máthéné Szigeti, László Tálas, Adrienn Széles, Zoltán Hargitai, Zsolt László Nagy, Miklós Nagy, Alexandra Kiss, Sándor Kéki, Gábor Szemán-Nagy

**Affiliations:** 1Department of Molecular Biotechnology and Microbiology, Faculty of Science, University of Debrecen, 4010 Debrecen, Hungary; 2Department of Lang Utilisation, Engineering and Precision Farming Technology, Faculty of Agricultural and Food Sciences and Environmental Management, University of Debrecen, Böszörményi 138, 4032 Debrecen, Hungary; 3Department of Pathology, Faculty of General Medicine, University of Debrecen, 4031 Debrecen, Hungary; 4Department of Applied Chemistry, Faculty of Science, University of Debrecen, 4010 Debrecen, Hungary; 5Institute of Chemistry, University of Miskolc, Miskolc-Egyetemváros, 3515 Miskolc, Hungary

**Keywords:** isocyanide, antifungal effect, *Aspergillus fumigatus*, invasive pulmonary aspergillosis murine model

## Abstract

As the recent outbreak of coronavirus disease 2019 (COVID-19) has shown, viral infections are prone to secondary complications like invasive aspergillosis with a high mortality rate, and therefore the development of novel, effective antifungals is of paramount importance. We have previously demonstrated that 1-amino-5-isocyanonaphthalene (ICAN) derivatives are promising original drug candidates against *Candida* strains (Patent pending), even against fluconazole resistant *C. albicans.* Consequently, in this study ICANs were tested on *Aspergillus fumigatus*, an opportunistic pathogen, which is the leading cause of invasive and systematic pulmonary aspergillosis in immunosuppressed, transplanted and cancer- or COVID-19 treated patients. We have tested several N-alkylated ICANs, a well as 1,5-naphthalene-diisocyanide (DIN) with the microdilution method against *Aspergillus fumigatus* strains. The results revealed that the diisocyanide (DIN) was the most effective with a minimum inhibitory concentration (MIC) value as low as 0.6 µg mL^−1^ (3.4 µM); however, its practical applicability is limited by its poor water solubility, which needs to be overcome by proper formulation. The other alkylated derivatives also have in vitro and in vivo anti-*Aspergillus fumigatus* effects. For animal experiments the second most effective derivative 1-N, N-dimethylamino-5-isocyanonaphthalene (DIMICAN, MIC: 7–8 µg mL^−1^, 36–41 µM) was selected, toxicity tests were made with mice, and then the antifungal effect of DIMICAN was tested in a neutropenic aspergillosis murine model. Compared to amphotericin B (AMB), a well-known antifungal, the antifungal effect of DIMICAN in vivo turned out to be much better (40% vs. 90% survival after eight days), indicating its potential as a clinical drug candidate.

## 1. Introduction

*Aspergillus fumigatus* is an opportunistic pathogen [1,2,3], the main cause of invasive aspergillosis, a serious infection, and can be a major cause of mortality in immunocompromised patients, with 59% one-year survival for people among solid organ transplant recipients and 25% among stem cell transplant recipients [4,5]. In a systematic review of intensive care unit autopsy studies, aspergillosis was one of the top four most common diagnoses that likely lead to death [6]. Moreover, as Denning et al. in 2011 calculated in 2011, 2.5% of adults who have asthma also have allergic bronchopulmonary aspergillosis (ABPA), which is approximately 4.8 million people worldwide, of which an estimated 400,000 also have chronic pulmonary aspergillosis (CPA) [7].

Patients with viral infections may be more susceptible to microbial secondary infections which can complicate disease management strategies and result in adverse patient outcomes [8,9]. One study found that ~40% of patients with severe COVID-19 pneumonia were also infected with filamentous fungi from the genus *Aspergillus*, giving rise to COVID-19 associated pulmonary aspergillosis (CAPA) [10,11]. A 2021 study found that all four patients who were admitted to an intensive care unit (ICU) with ARDS due to critical COVID-19 had infection with *A. fumigatus* strains [12]. Investigation of the CAPA isolates revealed that they are similar to the reference strains Af293 and CEA17 [12].

For treatment of invasive fungal infection only few antifungal agents can be used (caspofungin and flucytosine) [13,14]. Besides high mortality, *A. fumigatus* is developing resistance [15] against common antifungals, such as triazoles. In CDC’s 2019 Antibiotic Resistance Threats Report [16], azole-resistant *A. fumigatus* is on the watch list since it has the potential to spread rapidly. Azole-resistant *A. fumigatus* infections are difficult to treat, and these patients are up to 33% more likely to die than patients with infections that can be treated with azoles [17]. The development of novel (original) antifungals is therefore of paramount importance.

To date, several groups of antifungals have been developed and used. The first widely used antifungal agent was amphotericin B (AMB) against different fungi (*Candida* species, *Aspergillus fumigatus* and *Mucorales* infections) [18]. AMB attaches to ergosterol in the plasma membrane of fungi and changes the integrity of the membrane [19,20]. Its disadvantages are problems with its solubility, stability, oral bioavailability and nephrotoxicity [21,22], while if associated with lipids, it has a less toxic effect. Flucytosine is a member of pyrimidine analogues, which have effects on nucleic acid synthesis. It is transported into the fungal cell and after serial phosphorylation, flucytosine produces fluorinated nucleotides that interfere with DNA and RNA synthesis of the cell [14]. It is effective against *Candida* species, *Aspergillus fumigatus* and some other molds. Triazoles have effects on 14-α demethylase, one of the most important enzymes of sterol synthesis in fungi. Fluconazole, voriconazole and posaconazole belong to triazoles. Fluconazole has no effect on *Aspergillus* species and *Mucorales*, but is effective against *Candida* species [23]. Voriconazole and posaconazole are effective against *Aspergillus* and *Candida* species [24,25,26,27].

Caspofungin, micafungin and anidulafungin are echinocandins, and inhibit the catalytic subunit of 1,3-β-D-glucan synthase [28], which plays an important role in cell division and cell growth [13]. These antimycotics are effective against *Candida* spp. and *Aspergillus fumigatus*. A murine model of disseminated aspergillosis is often used for testing for the susceptibility to antifungal agents [29], and another advantage of this model is the similarity to human disease [30]. In addition, this model is well-characterized and reproducible.

We recently reported that our new, original molecule, 1-amino-5-isocyanonaphthalene (ICAN) and its derivatives (Figure 1) exert excellent antifungal activity against a broad range of *Candida* species, even against azole resistant *C. krusei* [31]. Their application can have two advantages: On the one hand they are novel and have never before been used in antifungal treatment, therefore no resistance could be formed against them. On the other hand, they can be prepared from the very cheap and simple 1,5-diaminonaphthalene (DAN) using dicholorocarbene to convert one (or two) amino groups into isocyanide resulting ICAN, which subsequently can be alkylated on the free amino group. This simple and cheap preparation is expected to be scaled-up easily, resulting in an inexpensive compound that can be used as an antifungal agent or disinfectant, even in agriculture.

Given the above effects of ICAN, the next step of this research would be to test the relevant effects of its derivatives, especially DIMICAN. Therefore, the aims of this study were to test 1-N,N-dimethylamino-5-isocyanonaphthalene (DIMICAN) and 1-amino-5-isocyanonaphthalene derivatives (MICAN, EICAN, PICAN and DIN) for their antifungal effect on *Aspergillus fumigatus* in vitro and, among them, to test the most effective agent in vivo in the aspergillosis murine model. After earlier experiments on *Candida albicans*, we thought that these agents can be effective against *Aspergillus fumigatus*. Our hypothesis was that DIMICAN has an inhibitory effect in vitro and in vivo in the aspergillosis murine model.

## 2. Results and Discussion

### 2.1. Properties of the Amino-Isocyanonaphthalenes (ICANs)

The structures and names of the compounds in this study are presented in Figure 1. Our previous structure-effect relationship investigations [31] revealed that for good antifungal activity (against *Candida* strains), both the naphthalene core and the isocyano-group(s) must be present. The conversion of one of the electron-donating (D) amino groups of 1,5-diaminonaphthalene DAN (Figure 1) to an electron withdrawing (A) isocyanide group enhances the polarity of the 1-amino-5-isocyanonaphthalene (ICAN) molecule considerably. The isocyano group is a strong H-acceptor, while the amino group can be both an H-donor and an H-acceptor depending upon the number of H-atoms connected to the amino group. Therefore, by alkylating the NH_2_-group or converting it to isocyanide, the secondary interactions with the chitosan wall of the fungi are assumed to be well controlled. It should be noted, however, that this study focuses only on the 1,5-ICAN derivatives, whereas ICANs are easy to modify and even the slightest change in the relative substitution position of the amino and isocyano groups on the naphthalene ring can result in a completely different behavior [32].

### 2.2. In Vitro Susceptibility Testing

The effectiveness of different ICAN derivatives (MICAN, DIMICAN, EICAN, PICAN and DIN) (Figure 1 and Table 1) were compared in susceptibility studies. The efficacy of all derivatives was tested against three *Aspergillus fumigatus* strains (MYA 3627, ATCC204305 and Af293). Stock solutions (1 mg/mL in DMSO) of the antifungal agents were diluted further in RPMI 1640 (with L-glutamine but without bicarbonate) to reach the desired concentration range (0.3–100 µg mL^−1^). Minimum inhibitory concentrations (MICs) and minimum effective concentrations (MECs) were read visually after 48 h incubation at 37 °C. (MEC-is the lowest concentration of antimicrobial agent that leads to growth of small, rounded, compact hyphal forms as compared to the hyphal growth seen in the growth control well.) We evaluated the MEC with a stereomicroscope. In this study, 80% inhibitory (MIC_80_) values are reported. Experiments were repeated a minimum of three times. The effect of amphotericin B (AMB) (Duchefa Biochemie) was also tested on all strains and MIC values were always smaller than 2 µg mL^−1^. The Af 293 (ATCC MYA-4609) strain was the less sensitive for all ICAN derivatives.

Several conclusions can be drawn from the results of Table 1. First, it is evident that the smallest, most nonpolar and rigid diisocyanide derivative DIN was the most effective since it had the lowest MICs, 0.6 µg mL^−1^ (3.4 µM) for all *A. fumigatus* strains. This MIC is not much higher than that of AMB (2.2 µM). The second most effective was DIMICAN (MIC: 7–8 µg mL^−1^, 36–41 µM), being more than 10 times than that of DIN). These MIC values are in the concentration range reported by previous studies. DIMICAN is still a small molecule with no free H-atoms on the amino group, therefore it can only be an H-acceptor. Interestingly, when one methyl group and one H-atom are connected to the N of the amino group (MICAN), the MIC values increase slightly to 11–12 µg mL^−1^. However, the exchange of the methyl group to longer ethyl or propyl groups results in a huge decrease in antifungal effect (MIC > 100 µg mL^−1^), therefore EICAN and PICAN had no effect on any *Aspergillus fumigatus* strains. In summary, small size, nonpolar character and the lack of NH bonds are keys to the antifungal effect against *A. fumigatus* strains. These features may suggest a key-lock mechanism with a certain part of the fungal cell; however, revealing the specific mechanism is a complex task and is the topic of further study.

### 2.3. Mouse Model of Invasive Aspergillosis and Antifungal Therapy

Based on the results of susceptibility studies, DIN turned out to be the most effective, however its application in vivo is limited by its poor solubility in water due to the rigid, aromatic and nonpolar structure. Therefore, for in vivo studies, DIMICAN as the second most effective molecule was selected. In the model of acute invasive aspergillosis, neutropenic mice were infected intranasally with *A. fumigatus* Af293 (3.5 × 10^6^ conidia/mouse) to establish an acute infection. 24 h after infection with conidia of *A. fumigatus*, animals were treated with AMB (5 mg kg^−1^) (dissolved in DMSO, final concentration of DMSO was 1%) or with DIMICAN (1.25, 2.5 or 5 mg kg^−1^) dissolved in olive oil. It should be noted, however, that we did pilot studies with DMSO (final concentration 1%) and olive oil, as well. It also should be noted here that DMSO (solvent of DIMICAN) did not have any effect on infected animals. The results are presented in Figure 2. It should be noted here that all experiments were repeated three times and each point in Figure 2 represents the average of the three experiments.

As it is evident from the figure, without treatment, 90% of the animals died after eight days. When animals were treated with AMB, 50% survived after five days and the survival ratio dropped to 40% after seven days. When animals were treated with DIMICAN at 1.25 mg kg^−1^, it had the same effect on survival as AMB in 5 mg kg^−1^. It should be noted that the data points of AMB (5 mg kg^−1^) and DIMICAN (1.25 mg kg^−1^) perfectly overlap, which is why one can only see five curves in Figure 2 instead of six. In contrast, when we increased the dose of DIMICAN to 2.5 mg kg^−1^, more than 60% of animals lived for eight days, while in animals treated with DIMICAN in 5 mg kg^−1^, 90% lived for eight days (Figure 2). 

In summary, 90% of untreated control animals died within eight days. AMB had (5 mg kg^−1^) a smaller effect on infected animals. DIMICAN (1.25, 2.5 or 5 mg kg^−1^) prolonged the survival period of animals (62.5% and 90%) compared with the survival period of non-treated infected (10%) controls (*p* < 0.05). We found a significant difference between the effect of AMB and 5 mg kg^−1^ DMICAN, while there were no significant differences between the effect of AMB and DIMICAN in 1.25 or 2.5 mg kg^−1^ concentrations. At these concentrations DIMICAN did not have any toxic effects on animals. The LD_50_ values are presented in [31] and, in addition, histopathological investigations were carried out to study the possible effect of DIMICAN on different organs.

### 2.4. Histopathological Studies

Before clinical application, it is very important to study what alterations the fungi and/or the antifungal agent will cause. Fungal stains are often used by the surgical pathologist when hints of fungal infection are observed with the H&E stain. In histopathological studies we observed tissue sections after H&E, PAS or GMS staining. Periodic acid-Schiff (PAS) staining is a rapid, relatively simple and reliable method and is among the most frequently employed techniques in histology laboratories. The PAS stain will reliably demonstrate the polysaccharide-laden wall of most fungal organisms, with the exception of *Histoplasma capsulatum* and the Grocott’s silver (GMS) stain, which is probably the most widely used fungal stain.

Hyphae were visible in the lungs of infected animals and infected animals treated with amphotericin B (Figure 3A–F). We found more hyphae in infected untreated animals than in infected and AMB-treated animals (Figure 3A–D).

Hyphae could not be detected in the lung of infected and DIMICAN treated animals (Figure 3A–C). When animals were treated with 1.25 mg kg^−1^ DIMICAN, mild inflammation was observed in the lung, but hyphae were not visible (Figure 3E,F). When animals were treated with DIMICAN at higher concentration, no inflammation was detected (Figure 3F).

We studied the effect of DIMICAN on not-infected animals and we did not observe any behavioral changes. DIMICAN caused elevated glycogen accumulation in the liver, and the hepatocytes structure changed to foamy (Figure 4A–D). Later, the glycogen level decreased to normal levels and hepatocytes were similar to animals not treated with DIMICAN (Figure 4A–D). For the formulation of DIMICAN we used olive oil, which leads to glycogen accumulation in the liver (Figure 4A,B).

### 2.5. Materials and Methods

#### 2.5.1. Materials

1,5-diaminonaphthalene (DAN) was purchased from Sigma-Aldrich (Schnelldorf, Germany) and used as received. 1-amino-5-isocyanonaphthalene (ICAN), 1-N-methylamino-5-isocyanonaphthalene (MICAN), 1-N,N-dimethylamino-5-isocyanonaphthalene (DIMICAN) and 1,5-diisocyanonaphthalene (DIN) were synthesized and characterized as described previously [33,34]. The synthesis procedure for 1-N-ethylamino-5-isocyanonaphthalene (EICAN) and 1-N-propylamino-5-isocyanonaphthalene (PICAN) as described previously [31].

*Aspergillus fumigatus* Af293 was cultured at 37 °C on nitrate minimal medium (NMM) [35]. To obtain conidia for infections, *A. fumigatus* was grown for six days on a nitrate minimal medium. Subsequently, conidia were washed in 1 mL aliquots of 0.9% *w*/*v* NaCl, 0.01 *v*/*v*% Tween 80 solution [36], were centrifuged three times (1500 g, 4 min), and were re-suspended in the spore washing solution. Conidia suspensions were diluted 10-fold, and the conidia were counted using a Bürker chamber.

#### 2.5.2. Clinical Isolates and In Vitro Susceptibility Testing

*Aspergillus fumigatus* MICs were determined according to the CLSI M38-A2 (CLSI, 2008). A stock solution of AMB (Duchefa Biochemie) was prepared in DMSO, diluted with RPMI 1640 medium (with L-glutamine but without bicarbonate) (Sigma Aldrich., St. Louis, MO, USA), and buffered to pH 7.0 with 0.165 M morpholinopropanesulfonic acid (MOPS; Sigma Aldrich., St. Louis, MO, USA). The final concentration range used for ICAN derivatives (diluted with RPMI 1640 from 1mg/mL DMSO stock solutions) was 0.3–100 µg mL^−1^, while that of AMB was 0.313 to 16 µg mL^−1^.

Testing was performed in 96-well round-bottom microtiter plates. Conidia suspensions were prepared in RPMI 1640 medium and were adjusted to give a final inoculum concentration of 0.4 × 10^4^ to 4 × 10^4^ conidia/mL. The plates were incubated at 37 °C and were read visually after 48 h. The MIC of AMB was defined as the lowest concentration at which there was 100% inhibition of growth compared with that in the drug-free control well. AMB resistance was defined as an MIC of 2 µg/mL, and AMB susceptibility was defined as an MIC of 0.5 µg mL^−1^. MIC determination was repeated at least twice.

#### 2.5.3. Animal Care

Animal experiments were carried out in our Experimental Animal Facility (reg. num.: III/3.-KÁT./2015) under the supervision of the Animal Care Committee, University of Debrecen. The experimental protocol was approved by the Animal Care Committee (license number: 2/2014 DEMAB to co-author GN). Animal experiments conformed to the general guidelines of the European Community (86/609/EEC) and special guidelines of BSL2 (200/54/EC 16(1)). Every group (10 each) of mice was housed in PI plastic cages (425/135/120 mm, 573.75 cm^2^) with mesh covers according to the guidelines of 2010/ 63/EU. Animals were fed with pelleted mouse chow (Purina, LabDiet, St. Louis, MO, USA) and tap water ad libitum. Automated room illumination of 12 h light and 12 h dark cycles, and room temperatures between 22–25 °C were maintained.

#### 2.5.4. Mouse Model

We used eight–10 week-old female and male BALB/c mice for all experiments. Mice were immunosuppressed using 250 mg kg^−1^ intraperitoneal (ip) cyclophosphamide (Endoxan, Baxter, Halle, Germany) (three days prior to infection) and 150 mg kg^−1^ ip cyclophosphamide (one and four days after infection). After immunosuppression, animals received gentamicin prophylaxis (5 mg kg^−1^ body weight) to prevent the bacterial infection of mice. Mice were given food and water *ad libitum* and were monitored daily. The animals were maintained in accordance with the Guidelines for the Care and Use of Laboratory Animals, and the experiments were approved by the Animal Care Committee of the University of Debrecen, Debrecen, Hungary (permission No. 4/2017). The method of inoculation followed the protocol of Palicz et al. (2013) [37] by inoculating 3.5 × 10^6^ in 50 μL of a suspension of conidia (7 × 10^7^/mL) intranasally to anesthetized mice. Under these conditions the solution reaches lungs through normal breathing.

#### 2.5.5. Antifungal Therapy

Each group (control and treatment) consisted of 10 animals. AMB (Duchefa Biochemie), DMSO, and DIMICAN treatment in a 0.5-mL bolus was started 24 h after infection. The doses were 1 mg AMB and 1.25, 2.5 or 5 mg DIMICAN per kg body weight. Both antifungals were administered intraperitoneally once daily for five consecutive days. Experiments were repeated three times. In animal experiments, olive oil was used as a delivery vehicle instead of DMSO (final concentration was 1%), because DIMICAN is lipophilic and it is well dissolved in olive oil. The oil of plants can be used as an alternative vehicle in drug formulation [38,39].

A paired *t*-test was used to determine the significant differences in AMB and DIMICAN in an aspergillosis murine model.

#### 2.5.6. Histopathological Studies

Control uninfected mice, infected mice, infected mice treated with AMB and infected mice treated with DIMICAN were euthanized on days three and six after infection (*n* = 3 mice/time point), and the lung and liver were removed, fixed with 10% formalin and embedded in paraffin wax. Tissue sections were stained with hematoxylin eosin (HE), periodic acid-Schiff (PAS) or Grocott’s Methamine Silver (GMS) stain.

## 3. Conclusions

Original 1-amino-5-isocyanonaphthalene (ICAN) derivatives were tested as potential new antifungal agents against *Aspergillus fumigatus*. The results of an in vitro CLSI broth microdilution revealed a clear structure-effect relationship, that is, free NH bonds, as well as larger than methyl substituents on the amino group, substantially decrease the antifungal effect against *A. fumigatus*. DIMICAN and DIN were the most active antifungal agents, while EICAN and PICAN had virtually no effect on fungal growth. In in vivo studies, infected animals were treated with DIMICAN, and its effect was compared to that of AMB. DIMICAN at three different concentrations (1.25, 2.5 and 5 mg kg^−1^) was applied in immunotherapy on five consecutive days 24 h after infection in an aspergillosis murine model. Based on survival period studies, it was found that DIMICAN at 1.25 mg kg^−1^ was as effective as AMB at 5 mg kg^−1^, while at 2.5 or 5 mg kg^−1^ it was more effective than AMB (90% vs. 40% survival after eight days). According to histopathological studies, aspergilloma was visible (in sections of lung) in both non-treated and AMB treated infected animals, however aspergilloma was much smaller in the AMB treated group. In DIMICAN treated animals aspergilloma could not be detected, while mild inflammation was easily observed. Eight days after DIMICAN treatment of uninfected animals, the liver was found in the same condition as in the control animals, indicating the possible clinical applicability of this derivative.

In the future, we plan to study the susceptibility of the antifungal effect of ICAN derivatives on other fungi (*Fusarium* species, *Cryptococcus neoformans*). We would like to observe its effect in vitro and in vivo.

## Figures and Tables

**Figure 1 jof-08-00985-f001:**
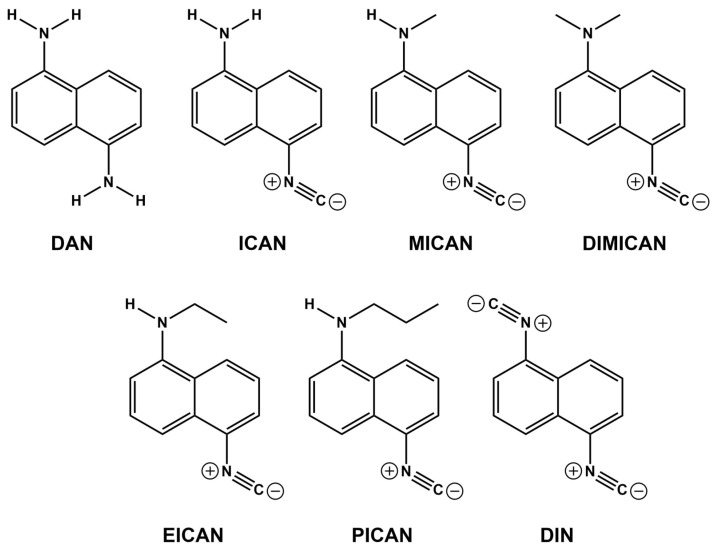
The structures and names of the compounds used in this study. 1,5-diaminonaphthalene (DAN), 1-amino-5-isocyanonaphthalene (ICAN), 1-N-methylamino-5-isocyanonaphthalene (MICAN) 1-N,N-dimethylamino-5-isocyanonaphthalene (DIMICAN), 1,5-diisocyanonaphthalene (DIN), 1-N-ethylamino-5-isocyanonaphthalene (EICAN) and 1-N-propylamino-5-isocyanonaphthalene (PICAN).

**Figure 2 jof-08-00985-f002:**
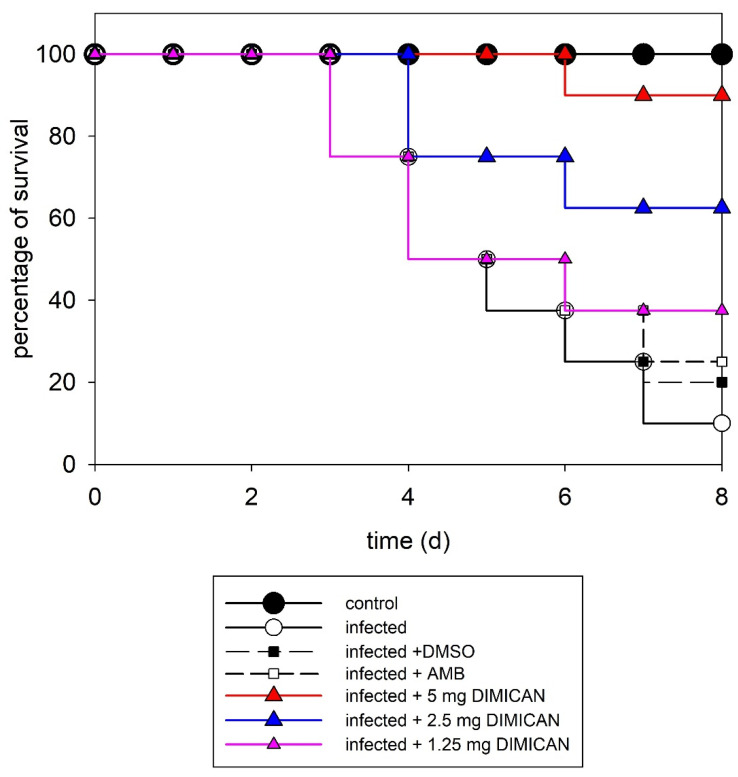
Survival curves for mice infected with *Aspergillus fumigatus* Af293. C- control, treated with DMSO, AMB or DIMICAN (D) in three different concentrations. For all groups *n* = 10. Number of independent experiments = 3.

**Figure 3 jof-08-00985-f003:**
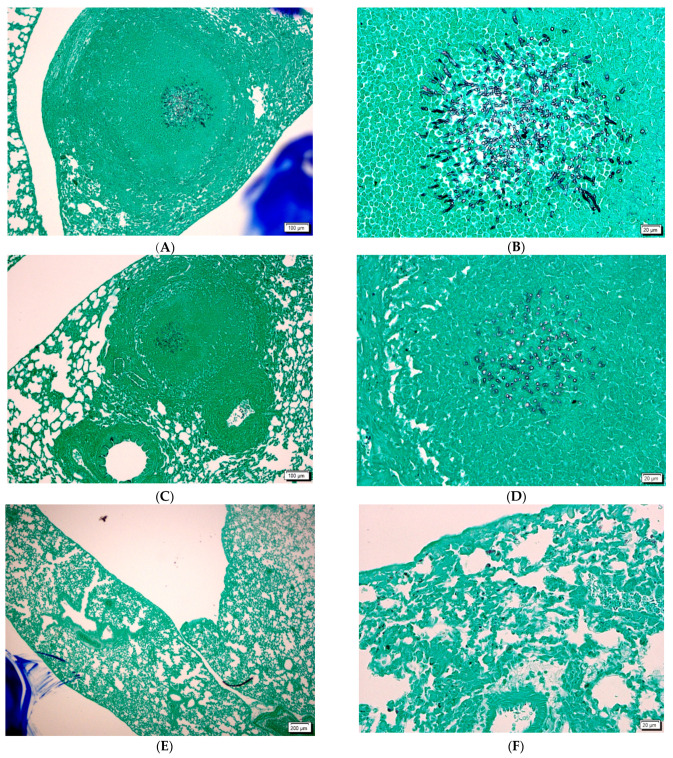
(**A**,**B**): aspergilloma in the lung of infected animal with GMS stain. (**C**,**D**): aspergilloma in the lung of infected and AMB treated animal with GMS stain. (**E**,**F**) lung of infected and DIMICAN treated animals with GMS strain. Scale bars (**A**,**C**,**E**) are 200 µm. Scale bars (**B**,**D**,**F**) are 20 µm.

**Figure 4 jof-08-00985-f004:**
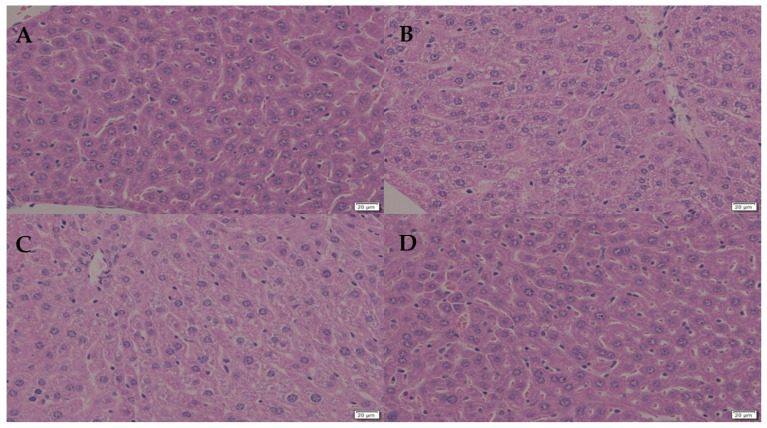
(**A**): control. (**B**): the effect of olive oil on the hepatocytes. (**C**): the effect of DIMICAN is visible in hepatocytes in not infected animals. (**D**): Ten days after DIMICAN treatment (**A**) hepatocytes were similar as control (**A**). Scale bars are 20 µm.

**Table 1 jof-08-00985-t001:** In vitro susceptibilities of *Aspergillus fumigatus* to MICAN, DIMICAN, EICAN, PICAN and DIN as determined by CLSI broth microdilution method. Minimum effective concentration (MEC) and 80% inhibitory (MIC_80_) values are presented. The values in parentheses belong to the molar concentration of the compounds. The best and second-best MIC values are indicated in red and green, respectively.

Compound	*Aspergillus fumigatus* Strains	MEC (µg/mL)	MIC_80_ (µg/mL)
MICAN (182.23 g/mol) ^a^	*A. fumigatus* MYA-3627	**9** (49 µM)	**11** (60 µM)
	*A. fumigatus* ATCC 204305	**9** (49 µM)	**11** (60 µM)
	*A. fumigatus* Af293	**10** (55 µM)	**12** (66 µM)
DIMICAN (196.25 g/mol) ^a^	*A. fumigatus* MYA-3627	**5** (25 µM)	**7** (36 µM)
	*A. fumigatus* ATCC 204305	**5** (25 µM)	**7** (36 µM)
	*A. fumigatus* Af293	**6** (30 µM)	**8** (41 µM)
EICAN (196.25 g/mol) ^a^	*A. fumigatus* MYA-3627	**50** (250 µM)	**100** (500 µM)
	*A. fumigatus* ATCC 204305	**50** (250 µM)	**100** (500 µM)
	*A. fumigatus* Af293	**50** (250 µM)	**100** (500 µM)
PICAN (210.28 g/mol) ^a^	*A. fumigatus* MYA-3627	**50** (240 µM)	**100** (480 µM)
	*A. fumigatus* ATCC 204305	**50** (240 µM)	**100** (480 µM)
	*A. fumigatus* Af293	**50** (240 µM)	**100** (480 µM)
DIN (178.19 g/mol) ^a^	*A. fumigatus* MYA-3627	**0.3** (1.7 µM)	**0.6** (3.4 µM)
	*A. fumigatus* ATCC 204305	**0.3** (1.7 µM)	**0.6** (3.4 µM)
	*A. fumigatus* Af293	**0.3** (1.7 µM)	**0.6** (3.4 µM)
AMB (924.08 g/mol) ^a^	*A. fumigatus* MYA-3627	<**2** (2.2 µM)	<**2** (2.2 µM)
	*A. fumigatus* ATCC 204305	<**2** (2.2 µM)	<**2** (2.2 µM)
	*A. fumigatus* Af293	<**2** (2.2 µM)	<**2** (2.2 µM)

^a^ Molecular weight (MW) of the compounds.

## Data Availability

Data is available upon request from the corresponding authors.

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
