# Peer review of "Potential Original Drug for Aspergillosis: In Vitro and In Vivo Effects of 1-N,N-Dimethylamino-5-Isocyanonaphthalene (DIMICAN) on Aspergillus fumigatus"

_jof, 2022, doi:10.3390/jof8100985_

Round 1

Reviewer 1 Report

Comments are shown with boxes on the original paper.

Reviewer 2 Report

In this ms., Szigeti et al. present results of their studies on the action of 1-N,N-dimethylamino-5-isocyanonaphthalene (DIMICAN) on Aspergillus fumigatus under in  vitro and in vivo (mouse model of disseminated aspergillosis) conditions. This is one of the series of compounds, isocyano derivatives of naphtalene, proposed in 2020 as antifungal drug candidates, with a broad spectrum of activity against human pathogenic yeasts of Candida spp. (ref. 31). In the present ms., a subject of interest is an anti-aspergillus activity of the same compounds. Neverthells, the compounds look interesting, their antifungal activity is promising. However, the present ms. should be modified before final acceptance. It cannot be excluded that repetition of some experiments is necessary.

Particular comments

1. The authors claim that they used Amphotericin B (AmB) as a reference antifungal in MIC determinations and in chemotherapy in the mouse model of disseminated aspergillosis. However, it is not known what kind of AmB preparation was actually used (no information in M&M section). AmB alone is very poorly soluble in aqueous solutions (usually less than 5 microgram/mL), so that at 5 microgram/mL or more it does not form true solutions and AmB precipitates formed may be a reason for mortality of model animals. On the other hand, some lipidic preparations of AmB, of which the most popular is AmB deoxycholate (Fungizone), are much better soluble, but in Fungizone, AmB constitutes about 50% of it, so that this must be taken into account in calculations of drug concentration.

It is clear therefore, that if the authors use AmB alone, they should repeat their experiments under conditions ensuring drug solubility. Definitely, 5 mg/kg b.w. in chemotherapy is too much. If Fungizone or any other lipid formulation of AmB was used, corrections for actual drug concentration should be done.

By the way, AmB or Fungizone does not seem to be the best choice as a reference, especially in chemotherapy tests. This is not the drug of choice in chemotherapy of aspergilloses. Instead, any candin, mainly anidulafungin or caspofungin, should be better used.

2. The authors have not made any attempt to reveal a possible mechanism of antifungal action of their compounds.  Instead, they speculate that the isocyano compounds could interact with components of the fungal cell wall: "These features may suggest a key-lock mechanism with a certain part of the fungal cell, most probably with the chitosan in the cell wall, since chitosan has a lot of strong H-bond donating free OH and NH2 groups."

First of all, the authors do not explain, how any possible interaction of DIMICAN (H-bond formation?) with fungal cell wall could result in antifungal effect. More importantly, the suggested site of interaction (chitosan) makes no sense. There are very few fungi containing chitosan in the cell wall, like Mucor rouxii. Most fungi, including Aspergillus spp. contain chitin, i.e. the fully N-acetylated derivative of chitosan. Obviously, formation of hydrogen bonds by DIMICAN with N-acetyl groups of chitin is possible but any growth inhibitory consequences of them seem highly unlikely.

In my opinion, any speculations about mechanism of the observed antifungal effect of DIMICAN are not justified. I  would suggest the authors to start extensive studies on the mechanism of action of this compound, focused on identification of an actual target. In my opinion (of course, also only a speculation), this is of intracellular character.

3. The survival curves (Figure 2) should be rather presented as a stairs-like graphs, not a polygonal chain.

4. The revised version of this ms, should be carefully corrected, In the original version there are many places where probably punctuation marks (comas, full stops) are missing,

Round 2

Reviewer 1 Report

Paper needs minor revision

Author Response

Response to each point of the comments of the Reviewers

Thank you for allowing us to submit a revised version of our manuscript titled Potential original drug for aspergillosis? In vitro and in vivo effects of 1-N,N-dimethylamino-5-isocyanonaphthalene (DIMICAN) on Aspergillus fumigatus to Journal of Fungi (article reference number: jof-1839226) We are grateful for the time and effort that you and the Reviewers have dedicated to advancing our manuscript. In the following, we give our point-by-point answers to the questions and comments of the Reviewers and the changes were tracked in the manuscript, too.

The authors are very grateful for the valuable remarks of Reviewer #1.

  1. “Please recheck the reference and species name.”

The Reference (31) is correct, and the species name was corrected to Candida krusei.

  1. “Please add this sentence in your manuscript also.”

This sentence was added to the manuscript.

“In addition, you mentioned above 80% inhibition?”

Yes, we know, 80 % inhibitory and MIC80 is not the same. we mentioned above 80% inhibition in all cases.

https://www.sciencedirect.com/science/article/pii/S0223523422000630

https://www.sciencedirect.com/science/article/pii/S0223523419302314

  1. “If so, you had 3x70=210 animals??? This is so much, is not it?

No it is not too much. We need data of all of these animals, that are necessary to get convincing results.

Yours sincerely,

                                                                                Zsuzsa Máthéné Szigeti, PhD